# Antibacterial Activity of Sulfated Galactans from *Eucheuma serra* and *Gracilari verrucosa* against Diarrheagenic *Escherichia coli* via the Disruption of the Cell Membrane Structure

**DOI:** 10.3390/md18080397

**Published:** 2020-07-29

**Authors:** Yixiang Liu, Yu Ma, Zhaohua Chen, Donghui Li, Wenqiang Liu, Ling Huang, Chao Zou, Min-Jie Cao, Guang-Ming Liu, Yanbo Wang

**Affiliations:** 1College of Food and Biological Engineering, Jimei University, Xiamen 361021, Fujian, China; my358108@163.com (Y.M.); zhchen@jmu.edu.cn (Z.C.); dh15589515872@163.com (D.L.); lwq19950512@163.com (W.L.); 15574291255@163.com (L.H.); zouchaoknight@163.com (C.Z.); mjcao@jmu.edu.cn (M.-J.C.); gmliu@jmu.edu.cn (G.-M.L.); 2Xiamen Key Laboratory of Marine Functional Food, Xiamen 361021, Fujian, China; 3National & Local Joint Engineering Research Center of Deep Processing Technology for Aquatic Products, Xiamen 361021, Fujian, China; 4School of Food Science and Biotechnology, Zhejiang Gongshang University, Hangzhou 310018, Zhejiang, China; wangyb@mail.zjgsu.edu.cn

**Keywords:** sulfated galactans, enterotoxigenic *Escherichia coli*, antibacterial activity, bacterial diarrhea, marine algae

## Abstract

Seaweed sulfated polysaccharides have attracted significant attention due to their antibacterial activity. This work investigated the antibacterial activity and mechanism of depolymerized sulfated galactans from *Eucheuma serra* (*E. serra*) and *Gracilaria verrucosa* (*G. verrucosa*) against enterotoxigenic *Escherichia coli* (ETEC) K88. The results show that removing the metal ions improves the anti-ETEC K88 activity of the galactans. The fluorescence labeling study confirmed that the sulfated galactans penetrated the cell walls and eventually reached the interior of the ETEC K88. Nucleic acid staining and intracellular protein leakage were also observed, indicating the destruction of permeability and integrity of the cell membrane. Interestingly, the two polysaccharides exhibited no effect on the proliferation of the selected Gram-positive bacteria and yeast. This indicates that the cell wall structure of the microorganisms could influence the bacteriostatic activity of the sulfated polysaccharides, as well. These results suggest that the sulfated seaweed polysaccharides might have potential application value in antibacterial diarrhea.

## 1. Introduction

Enterotoxigenic *Escherichia coli* (ETEC) is a kind of Gram-negative bacterium that can express a variety of membrane mucins, adhere to intestinal epithelial cells, and secrete enterotoxin to cause vomiting and diarrhea [1]. This pathogen is responsible for almost 600,000 deaths every year in most underdeveloped countries and regions, and mainly include children under 5 years old [2]. In addition, ETEC is also the leading cause of diarrhea in piglets and weaned piglets, causing substantial economic losses in the pig industry [3]. At present, antibiotics remain the primary treatment for ETEC-induced diarrhea, and although this type of treatment is extremely effective and inexpensive, the long-term use of antibiotics causes bacterial resistance. Since antibiotics can also damage the intestinal probiotics, leading to bacterial flora disorders in the gut [4], consumers are gradually turning their attention to natural active substances.

Polysaccharides from marine algae display a variety of potential pharmacological benefits to humans, such as anti-tumor, anti-inflammatory, antioxidant, and antiviral properties [5,6,7,8]. More recently, the bacteriostatic or bactericidal activities of sulfated seaweed polysaccharides have attracted increasing attention from consumers [9]. Previous studies indicate that fucoidan, a kind of sulfated polysaccharide from *Sargassum wightii* and *Laminaria japonica*, can effectively inhibit the growth of human pathogenic bacteria, such as *Vibrio cholera*, *Salmonella typhi*, and *Escherichia coli* (*E. coli*) [10,11]. Furthermore, sulfated polysaccharides from *Chaetomorpha aerea* exhibit an inhibitory effect on *Staphylococcus aureus*, but not fungi [12], while it is believed that the sulfated polysaccharides from red algae, *Porphyra haitanensis* and *Gracilaria lemaneiformis,* even displayed anti-diarrheal activity in ETEC K88-infected mice [13].

The antibacterial mechanism of cationic polysaccharides with a positive charge has been studied extensively and is relatively clear. For example, chitooligosaccharides, carrying a large number of amino groups, can bind to the negatively charged cell wall of bacteria via electrostatic adsorption, leading to the destruction of membrane permeability [14]. Moreover, chitooligosaccharides can also enter the interior of bacteria, compete with positively charged nuclear proteins, destroy DNA chains, and combine with it to achieve a bacteriostatic effect [15]. Sulfated polysaccharides are obviously different from cationic ones and are known as polyanions. Furthermore, it seems that anionic polysaccharides are unlikely to bind to negatively charged microorganism surfaces through electrostatic adsorption. However, during the invasion of animal host cells by pathogenic microorganisms, the polysaccharide receptors on the bacterial surface can specifically bind to the heparin sulfate on the surfaces of these cells, denoting a type of anionic sulfated polysaccharide [16]. Consequently, it is possible that the sulfated polysaccharides specifically recognize the polysaccharide receptors on the bacterial surface. Therefore, it is hypothesized that the sulfated polysaccharides can also degrade the cell wall structure and enter the bacterial cells, resulting in the death of the bacteria.

*Eucheuma serra* (*E. serra*) and *Gracilaria verrucosa* (*G. verrucosa*) are seaweeds from the southeast coastal areas of China and are the primary raw materials for the production of carrageenan and agar productions, respectively [17]. Previous studies have found that depolymerized sulfated galactans from both *E. serra* and *G. verrucosa* exhibited an inhibitory effect on diarrheagenic ETEC K88 [18]. This study investigated how the anti-ETEC K88 activity of sulfated galactans is enhanced by metal ion desorption since algae are responsible for enriching seawater with metal ions. More importantly, to further understand the antibacterial mechanism of the sulfated polysaccharides, a fluorescence labeling study was employed to observe their penetration into the cell walls of ETEC K88. Additionally, the influence of the cell wall structure of the microorganisms on their bacteriostatic activity was also considered by comparing the inhibitory differences on Gram-positive bacteria and fungi.

## 2. Results and Discussion

### 2.1. Removing the Metal Ions from Sulfated Galactans

The electronegative properties of sulfated polysaccharides determine their strong adsorption capacity of metal ions in seawater [19]. Therefore, it was necessary to improve the biological activity of the sulfated polysaccharides via the desorption of metal ions. Figure 1A shows that, at the beginning, the conductivity of the first filtrate of the *E. serra* sulfated polysaccharide (ESP) and *G. verrucosa* sulfated polysaccharide (GSP) solutions were 3060 and 2587 µs/cm, respectively. The conductivity of the filtrate was close to zero after eight ultrafiltration (dilution ratio was 8) repetitions, which was, therefore, used during the subsequent experiments to remove the metal ions from the polysaccharides. Figure 1B indicates that the ash content in ESP and GSP remained high at 17.8% and 13.2%, respectively, after ultrafiltration in the absence of a metal ion chelating agent in the eluent. However, the ash content in polysaccharides significantly (*p* < 0.05) decreased when the added ethylenediaminetetraacetic acid (EDTA) concentration was increased. When the EDTA concentration reached 1 × 10^−2^ mol/L, the ash content in ESP and GSP decreased to 8.2% and 6.7%, respectively (Figure 1B). At the same time, the sulfate content of polysaccharides increased slightly after metal ion removal. As shown in Figure 1C, under the condition of 1 × 10^−2^ mol/L EDTA, the sulfate group contents of ESP and GSP were increased from 28.5 ± 0.6% and 14.6 ± 0.2% to 30.0 ± 0.6% and 15.4 ± 0.5%, respectively.

The changes of above metal elements in the polysaccharides before and after desorption were investigated and the results are shown in Table 1. The results show that the Mg^2+^ and Ca^2+^ content in ESP and GSP were relatively high, exceeding 5000 and 1500 mg/kg, respectively. In addition, metal elements, such as Fe, Mn, and Zn, were also discovered in the marine sulfated polysaccharides. Metal ion removal reduced the Mg^2+^ content in ESP and GSP by 48% and 39%, respectively, while a corresponding decrease of 806.4 and 610.2 mg/kg was evident in the Ca^2+^ content levels. Therefore, it was apparent that combining metal ion chelating agents with ultrafiltration technology could effectively remove metal ions from marine polysaccharides.

### 2.2. Depolymerization of Sulfated Galactans

After degradation, the microstructure of the sulfated polysaccharides was observed using scanning electron microscopy (SEM). As shown in Figure 2, both ESP and DSP displayed a dense, flat appearance. After high-temperature and high-pressure treatment, the microstructure of the polysaccharides showed a visible change where the flaky structure was broken into porous flocculent. In previous research, an increase in reducing sugar and a decrease in viscosity were evident after the sulfated polysaccharides were hydrolyzed in high-temperature and high-pressure conditions [18].

### 2.3. The Effect of Metal Ion Removal on Antimicrobial Activity

The antibacterial activity of the polysaccharides against ETEC K88 was determined using the plate smearing method and the liquid turbidity method. Figure 3A–D shows that no obvious bacteriostatic effect was observed in the unhydrolyzed sulfated polysaccharides before or after metal ion removal. However, when the sulfated galactans from *E. serra* and *G. verrucosa* were degraded, the number of ETEC-K88 colonies on the medium were significantly reduced. Furthermore, only depolymerized sulfated galactans with the molecular weight (MW) ≤ 6.0 kDa displayed inhibitory effect on ETEC-K88, which is consistent with the results of a previous study [11]. Interestingly, fewer ETEC-K88 colonies were present on the medium when the depolymerized polysaccharides were exposed to the desorption of metal ions. The effect of metal ion removal on the antibacterial activity of the depolymerized polysaccharides was further verified using the liquid turbidity method (Figure 3E,F). After incubation for 24 h, the absorbance of the bacterial suspension at 600 nm gradually decreased when the depolymerized polysaccharide concentration before and after metal ion removal increased from 2.0 to 10.0 mg/mL. Compared with depolymerized ESP (D-ESP) without metal ion removal, the lower (*p* < 0.05) turbidity of the bacterial suspension appeared in the desalinized group ranging from 6.0 to 10.0 mg/mL. For depolymerized GSP (D-GSP), the desalinized group also exhibited better (*p* < 0.05) antibacterial activity at 10.0 mg/mL. 

Table 2 shows that, after desalination, the minimal inhibitory concentration (MIC) and minimal bactericidal concentration (MBC) values of D-ESP decreased from 10.0 L to 8.0 mg/mL and from 25.0 to 12.5 mg/mL, respectively, while the MIC value of D-GSP further decreased from 12.5 to 10.0 mg/mL. According to previous studies, metal ion removal enhanced the antioxidant activity of the sulfated polysaccharides from *Ascophyllum nodosum*, while the inhibitory effect of tea polysaccharides on α-glucosidase was also higher [20,21]. In this study, it seemed that both sugar chain degradation and metal ion removal effectively improved the antibacterial activity of sulfated polysaccharides against ETEC K88.

### 2.4. Observation of the Polysaccharides Entering the Bacterial Cells

It is believed that low MW chitosan can quickly enter *E. coli* cells via electrostatic adsorption, destroying the cell membrane structure [22]. However, due to the electronegativity of the bacterial cell wall, it is not clear whether anionic polysaccharides such as sulfated polysaccharides can bind and pass through it. The fluorescent dye 2-aminoacridone (2-AMAC) itself cannot penetrate the bacterial cell walls but can bind to the reducing end of the polysaccharide by decreasing the amination reaction [23]. Consequently, in this work, 2-AMAC was selected as the fluorescent label probe to investigate whether D-ESP or D-GSP can enter the cell interior of ETEC K88. Figure 4 indicates that, after the fluorescence-labeled sulfated polysaccharides were co-cultured with ETEC-K88 for 30 min, a distinct green fluorescence was observed in the cell interior using a fluorescent microscope. This result confirms that, as an anionic polymer, sulfated polysaccharides could penetrate cell walls to enter the cell interior of bacteria.

### 2.5. The Effect of Depolymerized Sulfated Galactans on Cell Membrane Permeability

Based on the results obtained regarding the entrance of sulfated polysaccharides into the ETEC K88 bacterial cells, the changes in the permeability of the cell membrane were further investigated during subsequent experiments. As a kind of nucleic acid fluorescent probe, propidium iodide (PI) can enter the bacterial cell interior when the cell membrane is damaged to combine with DNA molecules, resulting in the cells of the bacteria fluorescing red when viewed under an optical microscope [24]. Therefore, this work employed a PI fluorescence probe to reflect the membrane permeability of ETEC-K88 after polysaccharide treatment. Figure 5 indicates that only a few red cells were observed in the control group (without polysaccharide treatment). However, after D-ESP and D-GSP treatments at one or two times the MIC concentration, many ETEC K88 cells showing red fluorescence were observed in the field of vision. The present result is similar to an earlier report that the fluorescence intensity of *E. coli* treated with a 2MIC concentration of anthocyanin exceeded that of the blank group [24]. At the same time, it was also obvious that, compared with the 1MIC concentration, more red cells were uncovered when a 2MIC concentration of polysaccharides was used. These results demonstrate that sulfated polysaccharides could destroy the cell membrane permeability of ETEC-K88.

### 2.6. Analysis of the Cell Membrane Integrity

When the integrity of the microbial cell membrane is destroyed, proteins, nucleic acids, and other large molecules are released from the cell [25]. Therefore, in this study, the effect of sulfated polysaccharides on the cell membrane integrity of ETEC K88 was determined by analyzing intracellular protein release. Figure 6 indicates that Lanes 1, 4, and 7 represent the culture samples of ETEC K88 after sterile water or polysaccharide treatment for 1 h, while no obvious protein bands were observed. When the polysaccharide treatment time was extended to 3 h, obvious protein bands at 35 KDa were evident in Lanes 5 and 8, compared with the control group (Lane 2). This molecular weight was consistent with that of membrane-bound lytic transglycosylase (Mlt), which is present in the plasma membrane and outer membrane of *E. coli* [26]. It is believed that when the Mlt is released from the intracellular to the extracellular, it can act as a lysozyme and hydrolyze the murein polymer of the bacterial cell wall [26]. However, the protein band at 40 KDa was also observed after the D-GSP treatment. Unfortunately, the possible information represented by the protein is still unclear. In addition, it is also necessary to further study why different sulfated polysaccharide treatment may lead to different protein leakage of ETEC K88. In conclusion, both the sulfated polysaccharides, namely D-ESP and D-GSP, caused significant destruction to the cell membrane of ETEC K88, leading to intracellular protein leakage. The results are consistent with the findings of earlier studies. Zhang et al. (2017) found that the soluble proteins in the *E. coli* culture increased after treating with polysaccharides from *Cordyceps cicadae* [27]. Previous studies also confirmed that the sulfated polysaccharides could cause an increase in nucleic acid levels of the ETEC K88 culture [11,18].

### 2.7. Sensitivity to Different Microorganisms

Due to the differences in the cell wall structure of different microorganisms, this study further compared the sensitivity of sulfated polysaccharides to Gram-positive bacteria (including three intestinal probiotics and *S. aureus* ATCC 29213) and yeast (*Saccharomyces cerevisiae* BY4741). Figure 7 shows that when exposed to MBC concentrations (12.5 mg/mL for D-ESP and 25.0 mg/mL for D-GSP), both D-ESP and D-GSP exhibited no apparent inhibitory effect on the growth of intestinal probiotics, *Lactobacillus rhamnosus* ATCC 53103, *Bacillus coagulans* ATCC 7050, and *Bacillus subtilis* CMCC 63501, as well as yeast *Saccharomyces cerevisiae* BY4741. However, the proliferation of ETEC K88 was completely suppressed for both D-ESP and D-GSP. This may be attributed to the different cell wall structures of the microorganisms. It is believed that the outer cell wall of Gram-negative bacteria contains lipoproteins and that the peptidoglycan layer is thinner than that of Gram-positive bacteria. However, not only do the cell walls of Gram-positive bacteria have a much thicker peptidoglycan layer, but also the presence of phosphoric acid causes Gram-positive bacteria to be more negatively charged than Gram-negative bacteria [28,29]. In yeast, polysaccharides find it difficult to bind to the polysaccharide receptors on the cell surface and to penetrate the cell wall due to the wall’s thickness and the deeply encapsulated membrane proteins by dextran [24]. The results reveal that both D-ESP and D-GSP could selectively inhibit the proliferation of ETEC K88 but not intestinal probiotics, indicating their potential for anti-diarrheal application.

## 3. Materials and Methods

### 3.1. Chemical Reagents and Materials

The metal ion standard solutions were purchased from the National Institute of Metrology (Beijing, China). Gelatin was purchased from the Sangon Biotech Corporation Ltd. (Shanghai, China). The brain heart infusion (BHI) was purchased from Shanghai Gaochuang Chemical Technology Ltd. (Shanghai, China). Deionized water was produced using a Milli-Q unit (Millipore, Bedford, MA, USA). Other analytical reagent-grade reactants were purchased from the China National Pharmaceutical Industry Corporation Ltd. (Shanghai, China).

### 3.2. The Isolation and Purification of Sulfated Galactans

The collection and pretreatment of red algae *E. serra* and *G. verrucosa* were performed according to the methods used in an earlier study [18]. The sulfated galactans from *E. serra* were extracted and isolated based on previous studies [18,30]. Briefly, 100 g of the dried *E. serra* powder were macerated in water at 50 °C for 4 h. Then, the *E. serra* syrup was filtered through filter cloth, concentrated to 1/4th of the original volume, cooled, and precipitated overnight with three volumes of ethanol at 4 °C. The precipitate was collected via centrifugation (5000 r/min, 10 min), washed three times with 75% ethanol, dehydrated, and lyophilized to obtain a dried crude ESP.

The sulfated polysaccharides were obtained from *G. verrucosa* as described previously [18,31]. Briefly, the *G. verrucosa* powder was extracted twice with 50 °C water at a mass volume ratio of 1:40 (*w/v*) for 4 h and filtered through a filter cloth. The combined extracts were concentrated to 1/4th of initial volume in a rotary evaporator at reduced pressure at 60 °C. Then, 95% (*v/v*) ethanol was added to the concentrated supernatants, while being constantly stirred to acquire a final concentration of 40% (*v/v*) ethanol. The solution was left overnight at 4 °C and centrifuged at 5000 r/min for 10 min. The supernatant was again added to 95% (*v/v*) ethanol to obtain a final concentration of 80% (*v/v*) ethanol and kept overnight at 4 °C. The polysaccharides were collected via centrifugation using the same values as above and washed three times with 75% (*v/v*) ethanol, dehydrated, and lyophilized to obtain a dried crude GSP.

The crude sulfated polysaccharides were dissolved in distilled water to deproteinize them using the Sevag method [11]. The polysaccharide solution and the Sevag reagent (chloroform/n-butanol 4:1, *v/v*) were mixed (polysaccharide solution/Sevag reagent 5:1, *v/v*) and thoroughly shaken for 30 min, after which it was centrifuged to remove the denatured proteins. This process was repeated five times. After the precipitate was collected and lyophilized, the crude polysaccharide powders were kept in a glass desiccator at room temperature until use.

### 3.3. The Desorption of Metal Ions from the Polysaccharides

Based on the ultrafiltration technology, EDTA disodium was selected as the chelating agent for the removal of the metal ions absorbed on the sulfated galactans [32]. The crude polysaccharides were weighed and dissolved in distilled water at a final concentration of 0.5% (*w/v*). Then, acetic acid was added to the polysaccharide solution, and the pH adjusted to 5.0, after which the EDTA disodium was added at different concentrations. The solution was stirred (1200 r/min) at room temperature for 30 min and filtered using 4 kDa ultrafiltration membrane. During the ultrafiltration process, the conductivity of the filtrate was measured. When the conductivity value stabilized, the ultrafiltration was ceased, and the concentrated polysaccharide solution was lyophilized. The content of the polysaccharide sulfate group was determined using the gelatin barium chloride method [33], while the ash content was established with the burning weighing method [34]. The content of different metal ions (Mg^2+^, Ca^2+^, Fe^2+^, Mn^2+^, and Zn^2+^) was detected with atomic absorption spectrophotometry [35].

### 3.4. The Depolymerization of the Sulfated Galactans

According to a previous report, a high-temperature and high-pressure technique were used to degrade the polysaccharides during this study [11]. The polysaccharides containing 30% (*w*/*w*) deionized water was hydrolyzed in an autoclave reactor (Shanghai Boxun Industry & Commerce Co. Ltd. Medical Equipment Factory, Shanghai, China) at 121 °C and 0.103 MPa for 40 min. Then, the depolymerized products were dissolved in deionized water and fractionated via ultrafiltration. The fractions with MW ≤ 6 kDa were collected, freeze-dried, and stored in a glass dryer at room temperature until use. The morphological changes of D-ESP (MW ≤ 6 kDa) and D-GSP (MW ≤ 6 kDa) were observed using SEM (S-4800, Hitachi Corporation of Japan, Tokyo, Japan).

### 3.5. Bacterial Strains and Culture Conditions

A standard strain of Gram-negative ETEC K88 (CN-3-321) was purchased from Beijing Biobw Biotechnology Co., Ltd. (Beijing, China). *Lactobacillus rhamnosus* ATCC 53,103 was obtained from Valio Ltd. (Helsinki, Finland). *Saccharomyces cerevisiae* BY4741 was provided by Euroscarf (Oberursel, Germany). *S. aureus* ATCC 29213, *Bacillus coagulans* ATCC 7050, and *Bacillus subtilis* CMCC 63,501 were purchased from the China Industrial Microbial Species Preservation and Management Center (Beijing, China). *Lactobacillus rhamnosus* ATCC 53,103 and *Bacillus coagulans* ATCC 7050 were cultured in MRS liquid medium. *S. aureus* ATCC 29213, *Bacillus subtilis* CMCC 63501, and ETEC K88 were cultured in LB liquid medium. *Saccharomyces cerevisiae* BY4741 was cultured in YPD liquid medium.

### 3.6. Antibacterial Assay

The antibacterial activity of D-ESP and D-GSP against ETEC K88 was determined using both the plate smearing method and the liquid turbidity method [36,37]. First, the aqueous polysaccharide solutions were added to the sterilized nutrient agar before solidification, to obtain the final polysaccharide concentration of 7.5 mg/mL. After cooling, 100.0 μL of the ETEC K88 suspension (10^6^ CFU/mL) were spread onto the surface of the polysaccharide samples containing nutrient agar. After incubation at 37 °C for 24 h, the number of colonies on the plate surfaces was counted. Sterilized water was used as the negative control and kanamycin (0.05 mg/mL) as the positive control. The susceptibility of ETEC K88 to the sulfated galactans was also validated using the liquid turbidity method. During the logarithmic growth period, the ETEC K88 was centrifuged, after which it was diluted to a concentration of 10^6^ CFU/mL with normal saline water. Then, 1.6 mL LB medium, 2.0 mL polysaccharide solution, and 400.0 μL ETEC K88 suspension were mixed in each tube. The final polysaccharide concentrations were 2.0, 4.0, 6.0, 8.0, and 10 mg/mL, respectively. After incubation at 37 °C for 24 h, the absorbance was obtained at 600 nm using a microplate reader (TECAN Infinite 200 PRO, Grödig, Austria).

### 3.7. MIC and MBC

The MIC and MBC of D-ESP and D-GSP were performed as described by the National Committee for Clinical Laboratory Standards [25], with slight modifications. The MIC was determined as the lowest polysaccharide concentration at which no bacterial growth was detected after an incubation period of 24 h, while the MBC represented the lowest of test polysaccharide concentration that showed no growth in the culture after incubation at 37 °C for 48 h [11]. The polysaccharides were dissolved in sterilized 0.85% NaCl saline at a concentration of 50 mg/mL. The serial polysaccharide dilutions (25.0, 12.5, 10.0, 8.0, 6.25, 3.13, and 1.56 mg/mL) were prepared for both MIC and MBC tests, which were performed in LB culture with an inoculum of about 10^6^ CFU/mL. A series of tube dilutions was incubated on a rotary shaker at 180 rpm/min for either 24 or 48 h at 37 °C. Kanamycin solution (0.05 mg/mL) was used as the positive control and sterilized 0.85% NaCl saline was used as the negative control.

### 3.8. Observation of the Sulfated Polysaccharides Entering the Bacterial Cells

#### 3.8.1. Fluorescent Labeling of the Sulfated Polysaccharides

The fluorescence-labeled sulfated polysaccharides were prepared using the reductive amination method [23]. Here, 2 mg of either D-ESP or D-GSP were weighed and dissolved in 200 µL of 1.0 mol/L NaCNBH_3_ solution in a centrifuge tube. Then, 200 µL of 5 g/L 2-AMAC acetic acid/DMSO (3/17, *v/v*) solution were added to the system. After reacting in a water bath at 90 °C in the dark for 30 min, the mixture was immediately stored at −20 °C to terminate the reaction. The free 2-AMAC in the polysaccharide mixture was extracted with a tetrahydrofuran solution, and the precipitates were collected after centrifugation (5000 r/min) at 4 °C for 15 min. This process was repeated at least four times until no 2-AMAC was detectable in the extraction solution. The 2-AMAC levels in the extraction solution were analyzed using a fluorescence microscope (1 × 51, Olympus Corporation, Tokyo, Japan) at 450 (excitation) and 520 nm (emission). The fluorescence-labeled polysaccharide solutions were lyophilized and stored at 4 °C in the dark.

#### 3.8.2. Fluorescence Microscope Observations

The fluorescence-labeled or fluorescence-free sulfated polysaccharides were added into the bacterial suspension of ETEC K88 at the MIC concentration. Then, the bacterial suspension was incubated at 37 °C in the dark for 30 min, after which it was centrifuged at 5000 rpm for 15 min, and the microorganisms were washed three times with pH 7.4 Phosphate-Buffered Saline (PBS). Then, 5 µL of the bacterial suspension were fixed on a microscope slide and observed using a fluorescence microscope (1 × 51, Olympus Corporation, Tokyo, Japan) at 450 (excitation) and 520 nm (emission).

### 3.9. Observation of the Cell Membrane Permeability

The fluorescent probe PI was used to detect the membrane integrity according to an earlier study [24]. The ETEC K88 bacterial solution in the logarithmic growth phase was centrifuged, and the obtained bacteria were washed twice with PBS (pH 7.4) and resuspended at about 10^8^ CFU/mL. Then, the polysaccharide samples were added at final concentrations of 1MIC and 2MIC. After incubation at 37 °C for 2 h, the bacterial cells were collected via centrifugation (5000 r/min × 15 min), and the cells were washed twice with PBS (pH 7.4). Then, the bacteria were resuspended in pH 7.4 PBS at 10^8^ CFU/mL. The fluorescent probe PI was added to the bacterial suspension at a final concentration of 20.0 µmol/L and left to react at 37 °C in the dark for 20 min. Then, 5.0 µL of the bacterial suspension were fixed on a microscope slide and examined using a fluorescence microscope (RVL-100-G, ECHO, San Diego, CA, USA) at 488 nm (excitation) and 640 nm (emission).

### 3.10. Leakage of Intracellular Protein

The permeability of the cell membrane was determined by detecting the intracellular protein leakage in the culture medium [27]. The ETEC K88 bacterial solution in the logarithmic growth period was centrifuged, and the obtained cells were washed twice with PBS (pH 7.4). The microbes were resuspended in pH 7.4 PBS, while the optical density of the suspension was adjusted to 2.0. Then, the polysaccharide samples were added at a 1MBC concentration, after which the bacterial suspensions were incubated at 37 °C for 1, 3, and 5 h. After centrifugation 5000 r/min for 15 min, the supernatants were collected, concentrated ten times, and analyzed using SDS-PAGE gel electrophoresis.

### 3.11. The Effect of Sulfated Polysaccharides on the Growth of Different Microorganisms

The sulfated polysaccharides (D-ESP and D-GSP) were added to the culture medium containing different microorganisms at the concentration of MBC to ETEC K88, including intestinal probiotics (*Lactobacillus rhamnosus* ATCC 53103, *Bacillus coagulans* ATCC 7050, and *Bacillus subtilis* CMCC 63501), *S. aureus* ATCC 29213, and *Saccharomyces cerevisiae* BY4741. After 16 h of co-culturing the polysaccharides with different microorganisms, their growth was examined at 2 h intervals by determining the absorbance of the culture medium at 600 nm.

### 3.12. Statistical Analysis

The statistical significance of the differences between the control and treatment groups was determined with one-way ANOVA using Origin version 8.0 (OriginLab Corporation, Northampton, MA, USA), followed by Tukey tests. A normality test showed that all the raw data displayed normal distribution, while a variance test indicated that all groups exhibited equal variance. Data are expressed as mean ± SD of at least three individual experiments, each performed in triplicate. *p* < 0.05 was considered statistically significant.

## 4. Conclusions

Two types of sulfated galactans, namely ESP and GSP, were obtained from *E. serra* and *G. verrucosa*, respectively. After metal ion desorption and depolymerization, the improved anti-ETEC K88 activity of D-ESP and D-GSP was revealed, confirming that these sulfated polysaccharides can penetrate the cell membranes of ETEC K88 and eventually reach the cell interior, leading to the destruction of cell membrane permeability and integrity. Furthermore, D-ESP and D-GSP can selectively inhibit the proliferation of ETEC K88, but not the selected Gram-positive bacteria and yeast. This study implies that the low-MW sulfated galactans from marine red algae may display potential application value for the treatment of bacterial diarrhea.

## Figures and Tables

**Figure 1 marinedrugs-18-00397-f001:**
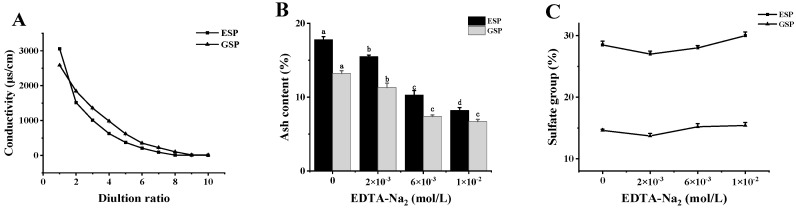
The effect of EDTA disodium combined with ultrafiltration technology on the desorption of metal ions in sulfated polysaccharides: (**A**) the conductivity changes in the ultrafiltration filtrate; (**B**) the influence of EDTA disodium on the ash content in ESP and GSP; and (**C**) the changes of sulfate group after metal ion removal. Data are expressed as mean ± SD (*n* = 3). Different letters indicate significant differences (*p* < 0.05).

**Figure 2 marinedrugs-18-00397-f002:**
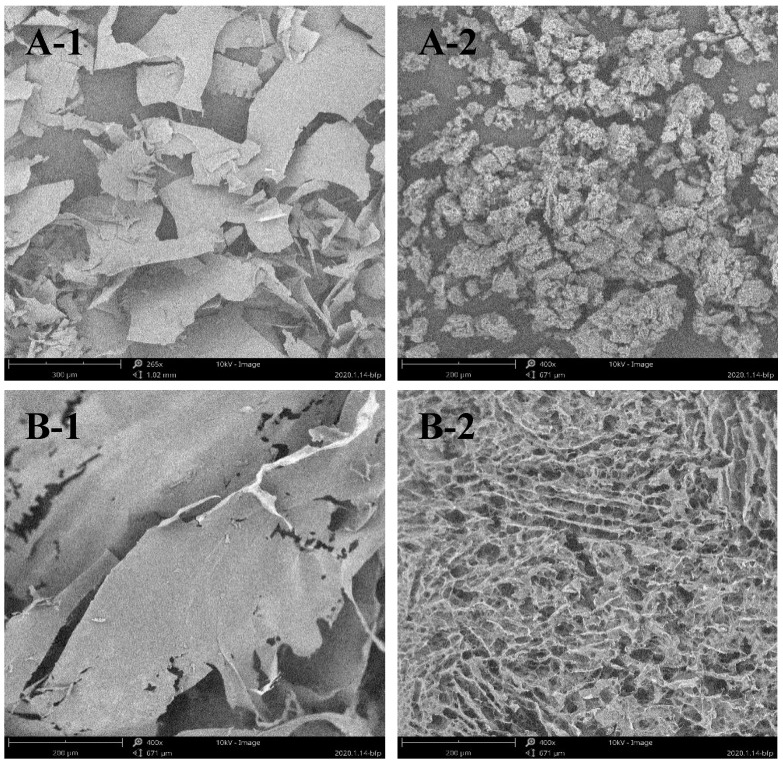
SEM observation of: the microstructure of the original long-chain sulfated polysaccharides (**A-1** and **B-1**); and the depolymerized sulfated polysaccharides (**A-2** and **B-2**).

**Figure 3 marinedrugs-18-00397-f003:**
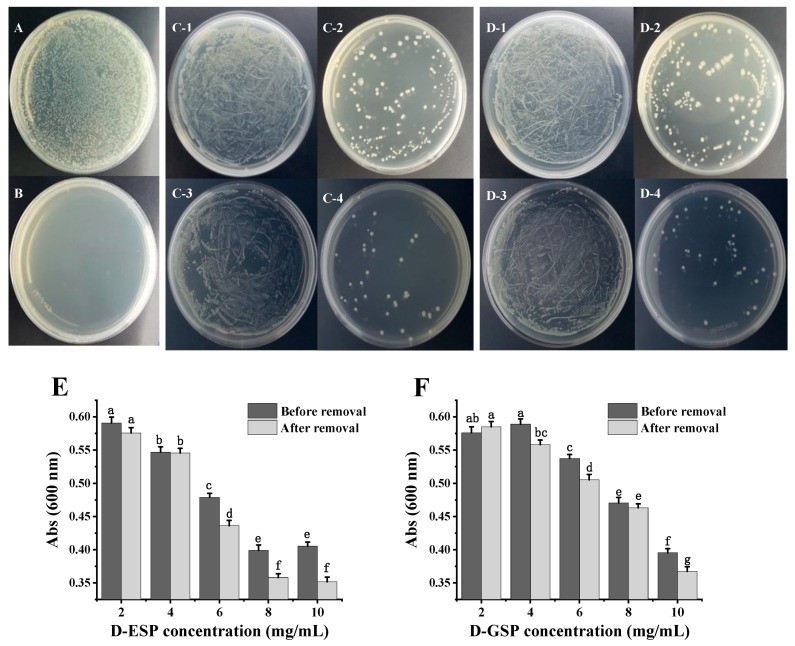
The antibacterial activity of sulfated polysaccharides against ETEC K88 based on the plate smearing method and liquid turbidity method: (**A**) the negative control (0.85% NaCl saline); (**B**) positive control (0.05 mg/mL kanamycin); (**C-1**) ESP and (**C-2**) D-ESP before metal ion removal; (**C-3**) ESP after metal ion removal; (**C-4**) D-ESP after metal ion removal; (**D-1**) GSP and (**D-2**) D-GSP before metal ion removal; (**D-3**) GSP after metal ion removal; (**D-4**) D-GSP after metal ion removal; (**E**) the antibacterial activity of D-ESP determined via the liquid turbidity method; and (**F**) the antibacterial activity of D-GSP determined with the liquid turbidity method. The polysaccharide concentration for the plate smearing method is 7.5 mg/mL. Data are expressed as mean ± SD (*n* = 3). Different letters indicate significant differences (*p* < 0.05).

**Figure 4 marinedrugs-18-00397-f004:**
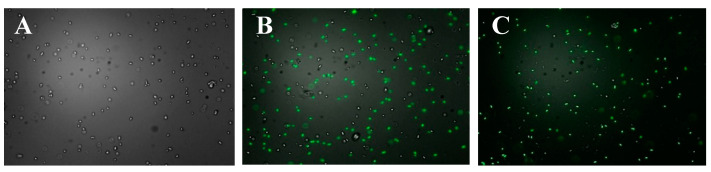
Observation of the D-ESP and D-GSP entering the cell interior of ETEC K88 using a fluorescent microscope (800×): (**A**) ETEC-K88 treated with 2-AMAC; (**B**) ETEC-K88 treated with 2-AMAC-labeled D-ESP; and (**C**) ETEC-K88 treated with 2-AMAC-labeled D-GSP.

**Figure 5 marinedrugs-18-00397-f005:**
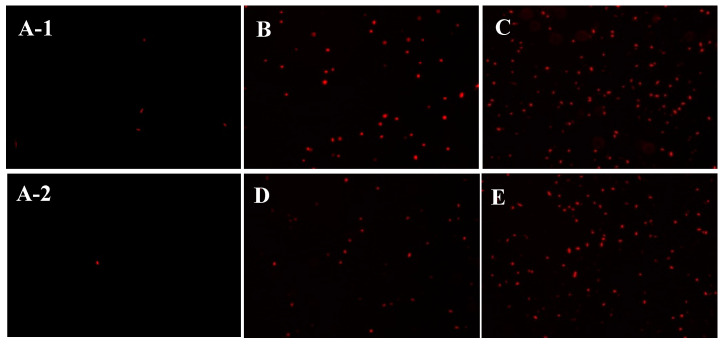
The effect of D-ESP and D-GSP on the cell membrane permeability of ETEC K88 under a fluorescent microscope (800×): (**A-1**,**A-2**) the control groups; (**B**) ETEC K88 treated with a 1MBC concentration of D-ESP; (**C**) ETEC K88 treated with a 2MBC concentration of D-ESP; (**D**) ETEC K88 treated with a 1MIC concentration of D-GSP; and (**E**) ETEC K88 treated with a 2MIC concentration of D-ESP.

**Figure 6 marinedrugs-18-00397-f006:**
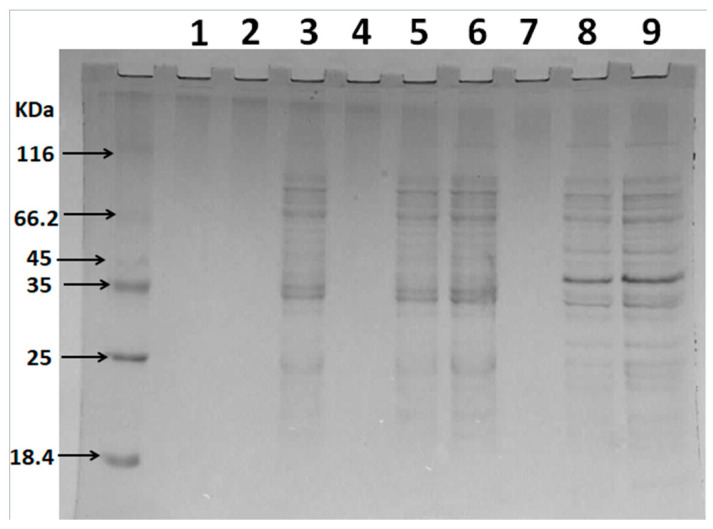
The effect of D-ESP and D-GSP on the intracellular protein leakage of ETEC-K88 based on SDS-PAGE analysis: Lanes 1–3, control groups (sterile water, 1, 3, and 5 h, respectively); Lanes 4–6, ETEC K88 treated with a 1MBC concentration of D-ESP for 1, 3, and 5 h respectively; and Lanes 7–9, ETEC K88 treated with a 1MBC concentration of D-GSP for 1, 3, and 5 h, respectively.

**Figure 7 marinedrugs-18-00397-f007:**
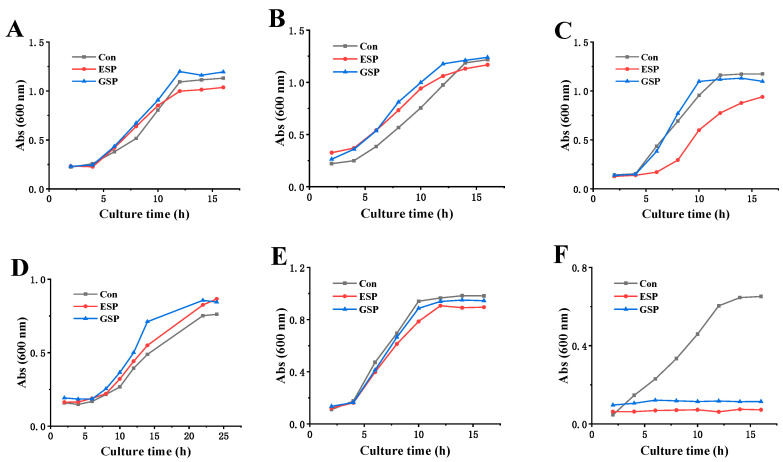
The effect of polysaccharide on the growth curve of intestinal bacteria: (**A**) *Lactobacillus rhamnosus* ATCC 53103; (**B**) *Bacillus coagulans* ATCC 7050; (**C**) *Bacillus subtilis* CMCC 63501; (**D**) *Saccharomyces cerevisiae* BY4741; (**E**) *S. aureus* ATCC 29213; and (**F**) ETEC K88.

**Table 1 marinedrugs-18-00397-t001:** The changes in the different metal elements in the polysaccharides after metal ion removal.

Items	Metal Elements (mg/kg)				
	Mg	Ca	Fe	Mn	Zn
ESP	5016.2 ± 78.2	4307.4 ± 49.6	356.7 ± 12.2	93.3 ± 2.2	36.7 ± 0.5
F-ESP	2643.7 ± 53.6	806.4 ± 25.4	234.9 ± 10.6	-	19.5 ± 0.3
GSP	5615.4 ± 81.3	1690.8 ± 36.1	79.1 ± 3.4	62.8 ± 1.3	33.2 ± 0.3
F-GSP	3477.7 ± 62.5	610.2 ± 21.3	41.7 ± 1.7	-	-

“-” means not detected; “F-ESP” means the filtered ESP by ultrafiltration; “F-GSP” means the filtered ESP by ultrafiltration.

**Table 2 marinedrugs-18-00397-t002:** The effect of metal ion removal on the antibacterial activity of sulfated polysaccharides.

Test Items	D-ESP		D-GSP	
	Before Removal	After Removal	Before Removal	After Removal
MIC (mg/mL)	10.0	8.0	12.5	10.0
MBC (mg/mL)	25	12.5	25.0	25.0

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
