# Peer review of "Antibacterial Activity of Sulfated Galactans from Eucheuma serra and Gracilari verrucosa against Diarrheagenic Escherichia coli via the Disruption of the Cell Membrane Structure"

_marinedrugs, 2020, doi:10.3390/md18080397_

Round 1

Reviewer 1 Report

see attachment

Reviewer 2 Report

It is a promising manuscript dealing with antibacterial activity against diarrheagenic Escherichia coli of sulfated galactans obtained from Eucheuma serra and Gracilari verrucose. The results described in the manuscript are exciting and potentially valuable; however, slightly discussed. Thus, authors should elaborate more discussion part. Also, scanning electron microscopy and/or transmission electron microscopy analysis required to show cell membrane damage caused by sulfated galactons. Please determine the content of nucleic acids and proteins in the supernatants and show the pattern of whole cell proteins of E. coli treated with sulfated galactans.

Additional comments:

L4: Diarrheagenic instead of Diarrheogenic

L36: ETEC and L71: E. serra and G. verrucosa (Please before use of abbreviations give the full term at first use).

L94: The low ash content indicating the low content of minerals. The authors analyzed the content of Mg, Ca, Fe, Mn, and Zn. Why they selected these elements? Please discuss the results.

L97: Figure 1 B (Description of statistics is missing).

L111: The saccharide composition determines the structure of galactans. Thus, the saccharide composition of depolymerized galactons should be presented for understating enhanced biological activity.

L127: What is the molecular weight of depolymerized galactons? ≤ 6 kDa? Please indicate.

L148: Figure 3 (Description of statistics is missing).

L190: Figure 5 (Please improve the figure quality).

L208: Please determine the content of nucleic acids and proteins in the supernatants and include the results.

L211: Figure 6 (SDS- pattern of whole-cell proteins of E. coli treated with sulfated galactans required to confirm the destruction of the cell membrane).

L279: Please present the results (the content of the polysaccharide sulfate group was determined using the gelatin barium chloride).

Round 2

Reviewer 2 Report

The authors answered the comments and improved the manuscript.